

# Comparison of Symbiodiniaceae diversities in different members of a *Palythoa* species complex (Cnidaria: Anthozoa: Zoantharia)—implications for ecological adaptations to different microhabitats

Masaru Mizuyama[1,2], Akira Iguchi[2], Mariko Iijima[2], Kodai Gibu[2,3] and James Davis Reimer[1,4]

[1] Molecular Invertebrate Systematics and Ecology Laboratory, Graduate School of Engineering and Science, University of the Ryukyus, Nishihara, Okinawa, Japan
[2] Geological Survey of Japan, National Institute of Advanced Industrial Science and Technology (AIST), Tsukuba, Ibaraki, Japan
[3] Department of Bioresources Engineering, National Institute of Technology, Okinawa College, Nago, Okinawa, Japan
[4] Tropical Biosphere Research Center, University of the Ryukyus, Nishihara, Okinawa, Japan

Corresponding author
Masaru Mizuyama,
mizuyama58@live.jp

## ABSTRACT

In this study we compared genotypes of zoantharian host-associating algal symbionts among *Palythoa* species, which are among the dominant benthic reef organisms in the Ryukyu Archipelago, Japan, and evaluated Symbiodiniaceae diversities of closely related congeneric *Palythoa* species. We targeted a species complex of the zoantharian genus *Palythoa* (*P. tuberculosa*, *P.* sp. yoron, *P. mutuki*) living among different microhabitats in a narrow reef area of Tokunoshima Island. For phylogenetic analyses, we used two DNA marker regions; nuclear internal transcribed spacer (ITS) and plastid mini-circle non-coding region (psbA$^{ncr}$), both of which have previously been used to determine Symbiodiniaceae genotypes of zoantharian species. Our results showed that all *Palythoa* species hosted symbionts of the genus *Cladocopium*, with genotypic compositions of this genus showing some variations among the three different *Palythoa* species. Additionally, we found that the *Cladocopium* genotypic composition was statistically different among *Palythoa* species, and among *P. tuberculosa* specimens in different microhabitats. Our results suggest that ecological divergence among these three *Palythoa* species may be related to differing Symbiodiniaceae diversities that may in turn contribute to eco-physiological adaptation into different microhabitats on coral reefs.

## INTRODUCTION

Zoantharians (Anthozoa: Zoantharia) belong to the phylum Cnidaria and can be dominant organisms in shallow coral reef areas (e.g., *Burnett et al., 1994*). In particular, the genus

*Palythoa* is often among the most dominant benthos in coral reef areas (*Irei, Nozawa & Reimer, 2011*; *Santos et al., 2016*; *Reimer et al., 2017a*).

We recently reported on four putative *Palythoa* species (*P. tuberculosa*, *P.* sp. yoron, *P. mutuki*, and *P.* aff. *mutuki*) that form a species complex, and were observed to all occur within a narrow range of coral reefs in southern Japan (*Mizuyama, Masucci & Reimer, 2018*). For example, *P. tuberculosa* tends to occur across a wide range of habitats from shallow to deeper areas, from the intertidal zone to the mesophotic reef slope (*Mizuyama, Masucci & Reimer, 2018*), and has been reported from tropical to temperate regions (*Reimer, Takishita & Maruyama, 2006*). On the other hand, the other three *Palythoa* species appear to more restricted compared to *P. tuberculosa* in terms of their distribution and habitats within coral reefs. *Palythoa mutuki* is the second most dominant species in this genus in Okinawa and is often dominant at the reef edge, in surge channels, and in small bumps on reef flats (*Irei, Nozawa & Reimer, 2011*). *Palythoa* sp. yoron has yet to be formally described, but tends to occur on reef flats and backreef moats where it is exposed to strong water currents (*Shiroma & Reimer, 2010*). Although there is little published information on *P.* aff. *mutuki*, it has been observed near *P. mutuki* colonies on the reef flat (*Mizuyama, Masucci & Reimer, 2018*). Although molecular delineation of these *Palythoa* species groups was unsuccessful with molecular data, likely due to incomplete lineage sorting, they can be distinguished via morphological and reproductive data (*Mizuyama, Masucci & Reimer, 2018*). In addition, these *Palythoa* species display different microhabitat patterns within the coral reef, but it is still unclear how these species would have diversified under almost completely sympatric conditions.

Symbiodiniaceae endosymbiotic dinoflagellates are symbiotic with various metazoan phyla including Cnidaria (*LaJeunesse et al., 2018*). Many zoantharians maintain Symbiodiniaceae, similar to reef-building corals (*Noda et al., 2017*; *Wee, Kurihara & Reimer, 2019*). In the case of scleractinian corals, symbiotic relationships with Symbiodiniaceae are important for host survival in various environments (*Baker, 2003*), and can contribute to ecological divergence of coral host species (*Winters et al., 2009*). Previous molecular studies have reported that species composition of Symbiodiniaceae is closely related to host genotypes in corals (e.g., *Bongaerts et al., 2010*; *Pinzon & LaJeunesse, 2011*). Thus, information on the composition Symbiodiniaceae of the four *Palythoa* species above would also be helpful to understand their ecological divergence into different microenvironments within a reef. In particular, genotypic composition of symbiotic algae would be informative for understanding ecological divergence of these species because the genetic and/or community changes of microbiomes are expected to be faster than that of the hosts themselves (*Torda et al., 2017*), facilitating eco-physiological adaptation of holobionts into different microenvironments (e.g., *Reimer et al., 2017b*; *Wee, Kurihara & Reimer, 2019*). In this study, we aimed to (1) compare diversities of symbionts among the closely related *Palythoa* species *P. tuberculosa*, *P.* sp. yoron, *P. mutuki* and *P.* aff. *mutuki*, and (2) determine if diversities of symbionts explain eco-physiological adaptations to microhabitats of each species that entailed divergences among them (*P. tuberculosa*, *P.* sp. yoron and *P. mutuki*).

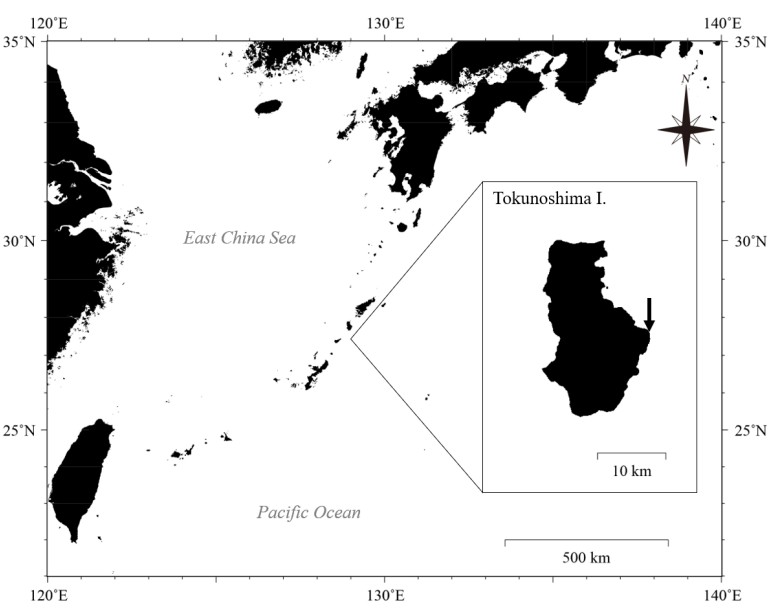

**Figure 1** **Location of Tokunoshima Island and the sampling site (arrow in inset) for the *Palythoa* specimens in this study.** Map data: GeoLite2 data created by MaxMind using the Generic Mapping Tools (GMT v5.4.5) software package. CC BY SA 4.0.

## MATERIALS & METHODS

### Specimens collection

Eighty-two colonies of three *Palythoa* species (*P. tuberculosa*, *P.* sp. yoron, and *P. mutuki*) were collected from a shallow fringing reef of Tokunoshima Island, Kagoshima, Japan (Figs. 1 and 2). Specimens of these three *Palythoa* species were collected in four different areas (Table 1, Fig. 2A): reef edge (Fig. 2B, 27.76998333N, 129.03988611E) for *P. tuberculosa* (Fig. 2C); reef flat 1 (Fig. 2D, 27.76997777N, 129.03925000E) for *P. tuberculosa* (Fig. 2E) and *P. mutuki*; reef flat 2 (Fig. 2F, 27.77195277N, 129.03843611E) for *P. mutuki* (Fig. 2G); and backreef moat (Fig. 2H, 27.76990833N, 129.03855833E) for *P. tuberculosa* and *P.* sp. yoron (Fig. 2I). To avoid collecting clones, we collected individuals from clearly different colonies while maintaining a set distance from each other of at least 1 m. In a previous study, even when closer to each other (within approximately 50 × 50 cm), no clones were observed in *Zoanthus* (Cnidaria: Anthozoa: Zoantharia) colonies (*Albinsky et al., 2018*). In addition, eighteen previously collected specimens of *Palythoa* species including 10 *P. aff. mutuki* specimens from *Mizuyama, Masucci & Reimer (2018)* were also examined in this study (Table 1).

### DNA extraction and PCR amplification

From each of these specimens, several polyps were cut with a surgical knife and DNA was extracted using DNeasy Blood and Tissue Kit (QIAGEN). DNA concentrations were checked by Qubit Fluorometer (ThermoFisher, Waltham, USA). Two molecular markers for genotyping symbiotic algae of *Palythoa* species were examined: nuclear internal

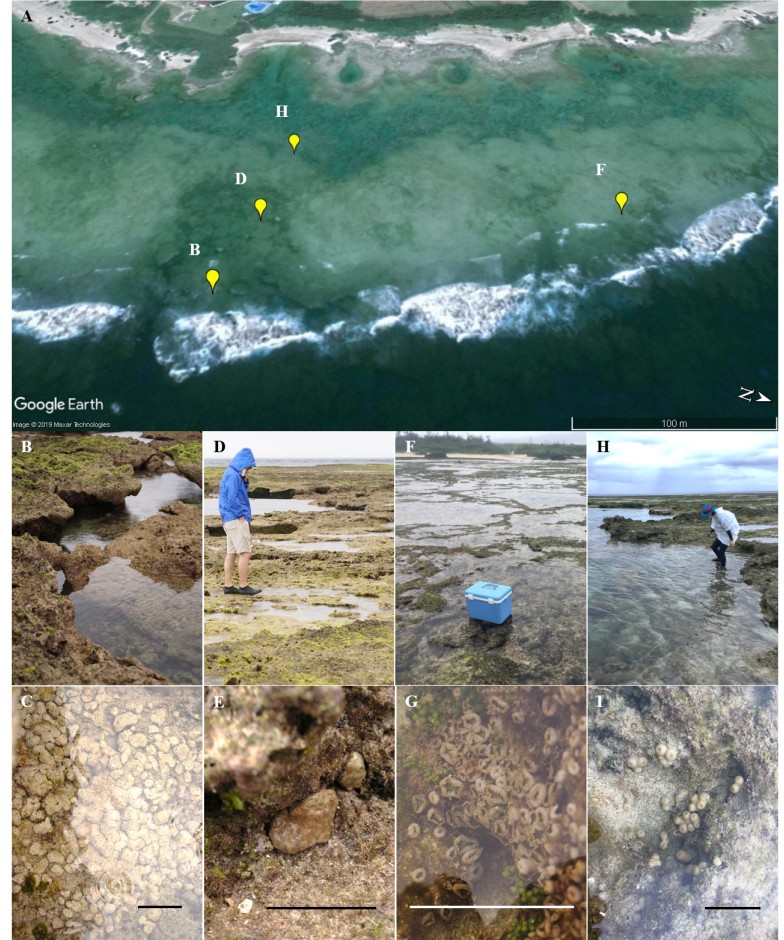

**Figure 2** **Landscape of the coral reef flat at the study site and in situ images of *Palythoa* species used in this study.** (A) Satellite image of the reef area obtained by Google Earth; (B) reef edge; (C) *P. tubeculosa*; (D) reef flat 1; (E) *P. tuberculosa*; (F) reef flat 2; (G) *P. mutuki*; (H) backreef moat; (I) *P.* sp. yoron. Map data: Google, Maxar Technologies. Scale bars in C, E, G, and I are 10 cm.

transcribed spacer ribosomal DNA (ITS-rDNA) region including partial 18S–ITS1–5.8S–ITS2–partial 28S (primers zITSf: CCG GTG AAT TAT TCG GAC TGA CGC AGT and ITS4: TCC TCC GCT TAT TGA TAT GC, (*Baillie, Belda-Baillie & Maruyama, 2000*; appx. 700–750 bp) and plastid mini-circle non-coding region DNA (psbA$^{ncr}$) (primers 7.4-Forw: GCA TGA AAG AAA TGC ACA CAA CTT CCC and 7.8-Rev: GGT TCT CTT ATT CCA TCA ATA TCT ACT G, (*Moore et al., 2003*; appx. 800–850 bp). These regions were amplified according to the PCR thermal conditions in *Wee, Kurihara & Reimer (2019)*. Amplified PCR products of symbionts were directly sequenced, and sequence data were manually checked based on the chromatogram files and low quality sites were removed at the 5′ and 3′ ends by BioEdit v.7.0.5.3 (*Hall, 1999*). Obtained sequences were deposited in the GenBank database (MN654128–MN654306, Table 1).

Mizuyama et al. (2020), *PeerJ*, DOI 10.7717/peerj.8449

**Table 1** Specimen list.

| Specimen ID | Location/Region | Spiecies ID | Date (m/d/y) | Environment | Accession no. of ITS | Accession no. of psbA-F | Accession no. of psbA-R |
|---|---|---|---|---|---|---|---|
| A01PtToKa | Kaminomine/Tokunoshima | *Palythoa tuberculosa* | Jun 2, 2019 | Reef edge | MN654209 | MN654185 | – |
| A02PtToKa | Kaminomine/Tokunoshima | *Palythoa tuberculosa* | Jun 2, 2019 | Reef edge | MN654210 | MN654184 | MN654134 |
| A03PtToKa | Kaminomine/Tokunoshima | *Palythoa tuberculosa* | Jun 2, 2019 | Reef edge | MN654211 | – | – |
| A04PtToKa | Kaminomine/Tokunoshima | *Palythoa tuberculosa* | Jun 2, 2019 | Reef edge | MN654212 | MN654186 | MN654135 |
| A05PtToKa | Kaminomine/Tokunoshima | *Palythoa tuberculosa* | Jun 2, 2019 | Reef edge | MN654213 | MN654187 | MN654136 |
| A06PtToKa | Kaminomine/Tokunoshima | *Palythoa tuberculosa* | Jun 2, 2019 | Reef edge | MN654214 | MN654188 | – |
| A07PtToKa | Kaminomine/Tokunoshima | *Palythoa tuberculosa* | Jun 2, 2019 | Reef edge | MN654215 | MN654189 | MN654137 |
| A08PtToKa | Kaminomine/Tokunoshima | *Palythoa tuberculosa* | Jun 2, 2019 | Reef edge | MN654216 | MN654190 | MN654138 |
| A09PtToKa | Kaminomine/Tokunoshima | *Palythoa tuberculosa* | Jun 2, 2019 | Reef edge | MN654217 | – | – |
| A11PtToKa | Kaminomine/Tokunoshima | *Palythoa tuberculosa* | Jun 2, 2019 | Reef flat | MN654218 | MN654191 | MN654139 |
| A12PtToKa | Kaminomine/Tokunoshima | *Palythoa tuberculosa* | Jun 2, 2019 | Reef flat | MN654219 | MN654192 | MN654140 |
| A13PtToKa | Kaminomine/Tokunoshima | *Palythoa tuberculosa* | Jun 2, 2019 | Reef flat | MN654220 | MN654193 | – |
| A14PtToKa | Kaminomine/Tokunoshima | *Palythoa tuberculosa* | Jun 2, 2019 | Reef flat | MN654221 | – | – |
| A15PtToKa | Kaminomine/Tokunoshima | *Palythoa tuberculosa* | Jun 2, 2019 | Reef flat | MN654222 | – | – |
| A16PtToKa | Kaminomine/Tokunoshima | *Palythoa tuberculosa* | Jun 2, 2019 | Reef flat | MN654223 | MN654194 | MN654141 |
| A17PtToKa | Kaminomine/Tokunoshima | *Palythoa tuberculosa* | Jun 2, 2019 | Reef flat | MN654224 | – | – |
| A18PtToKa | Kaminomine/Tokunoshima | *Palythoa tuberculosa* | Jun 2, 2019 | Reef flat | MN654225 | MN654195 | MN654142 |
| A19PtToKa | Kaminomine/Tokunoshima | *Palythoa tuberculosa* | Jun 2, 2019 | Reef flat | MN654226 | MN654198 | – |
| A20PtToKa | Kaminomine/Tokunoshima | *Palythoa tuberculosa* | Jun 2, 2019 | Reef flat | MN654227 | – | – |
| A21PtToKa | Kaminomine/Tokunoshima | *Palythoa tuberculosa* | Jun 2, 2019 | Backreef moat | MN654228 | MN654169 | MN654159 |
| A22PtToKa | Kaminomine/Tokunoshima | *Palythoa tuberculosa* | Jun 2, 2019 | Backreef moat | MN654229 | – | – |
| A24PtToKa | Kaminomine/Tokunoshima | *Palythoa tuberculosa* | Jun 2, 2019 | Backreef moat | MN654230 | – | MN654143 |
| A25PtToKa | Kaminomine/Tokunoshima | *Palythoa tuberculosa* | Jun 2, 2019 | Backreef moat | MN654231 | – | – |
| A26PtToKa | Kaminomine/Tokunoshima | *Palythoa tuberculosa* | Jun 2, 2019 | Backreef moat | MN654232 | – | – |
| A27PtToKa | Kaminomine/Tokunoshima | *Palythoa tuberculosa* | Jun 2, 2019 | Backreef moat | MN654233 | – | – |
| A28PtToKa | Kaminomine/Tokunoshima | *Palythoa tuberculosa* | Jun 2, 2019 | Backreef moat | MN654234 | – | – |
| A29PtToKa | Kaminomine/Tokunoshima | *Palythoa tuberculosa* | Jun 2, 2019 | Backreef moat | MN654235 | – | – |

**Table 1** (*continued*)

| Specimen ID | Location/Region | Spiecies ID | Date (m/d/y) | Environment | Accession no. of ITS | Accession no. of psbA-F | Accession no. of psbA-R |
|---|---|---|---|---|---|---|---|
| A30PtToKa | Kaminomine/Tokunoshima | *Palythoa tuberculosa* | Jun 2, 2019 | Backreef moat | MN654236 | – | – |
| B01PmToKa | Kaminomine/Tokunoshima | *Palythoa mutuki* | Jun 2, 2019 | Reef flat | MN654237 | – | – |
| B02PmToKa | Kaminomine/Tokunoshima | *Palythoa mutuki* | Jun 2, 2019 | Reef flat | MN654238 | MN654199 | MN654144 |
| B03PmToKa | Kaminomine/Tokunoshima | *Palythoa mutuki* | Jun 2, 2019 | Reef flat | MN654239 | – | – |
| B04PmToKa | Kaminomine/Tokunoshima | *Palythoa mutuki* | Jun 2, 2019 | Reef flat | MN654240 | – | – |
| B05PmToKa | Kaminomine/Tokunoshima | *Palythoa mutuki* | Jun 2, 2019 | Reef flat | MN654241 | – | MN654145 |
| B06PmToKa | Kaminomine/Tokunoshima | *Palythoa mutuki* | Jun 2, 2019 | Reef flat | MN654242 | MN654170 | MN654160 |
| B07PmToKa | Kaminomine/Tokunoshima | *Palythoa mutuki* | Jun 2, 2019 | Reef flat | MN654243 | – | MN654161 |
| B08PmToKa | Kaminomine/Tokunoshima | *Palythoa mutuki* | Jun 2, 2019 | Reef flat | MN654244 | MN654171 | MN654162 |
| B09PmToKa | Kaminomine/Tokunoshima | *Palythoa mutuki* | Jun 2, 2019 | Reef flat | MN654245 | – | – |
| B11PmToKa | Kaminomine/Tokunoshima | *Palythoa mutuki* | Jun 2, 2019 | Reef flat | MN654246 | – | – |
| B12PmToKa | Kaminomine/Tokunoshima | *Palythoa mutuki* | Jun 3, 2019 | Reef flat | MN654247 | MN654172 | – |
| B13PmToKa | Kaminomine/Tokunoshima | *Palythoa mutuki* | Jun 3, 2019 | Reef flat | MN654248 | – | – |
| B14PmToKa | Kaminomine/Tokunoshima | *Palythoa mutuki* | Jun 3, 2019 | Reef flat | MN654249 | MN654173 | – |
| B15PmToKa | Kaminomine/Tokunoshima | *Palythoa mutuki* | Jun 3, 2019 | Reef flat | MN654250 | – | – |
| B16PmToKa | Kaminomine/Tokunoshima | *Palythoa mutuki* | Jun 3, 2019 | Reef flat | MN654251 | – | – |
| B17PmToKa | Kaminomine/Tokunoshima | *Palythoa mutuki* | Jun 3, 2019 | Reef flat | MN654252 | MN654174 | MN654163 |
| B18PmToKa | Kaminomine/Tokunoshima | *Palythoa mutuki* | Jun 3, 2019 | Reef flat | MN654253 | MN654200 | – |
| B20PmToKa | Kaminomine/Tokunoshima | *Palythoa mutuki* | Jun 3, 2019 | Reef flat | MN654254 | – | – |
| B21PmToKa | Kaminomine/Tokunoshima | *Palythoa mutuki* | Jun 3, 2019 | Reef flat | MN654255 | – | – |
| B22PmToKa | Kaminomine/Tokunoshima | *Palythoa mutuki* | Jun 3, 2019 | Reef flat | MN654256 | – | – |
| B23PmToKa | Kaminomine/Tokunoshima | *Palythoa mutuki* | Jun 3, 2019 | Reef flat | MN654257 | – | – |
| B24PmToKa | Kaminomine/Tokunoshima | *Palythoa mutuki* | Jun 3, 2019 | Reef flat | MN654258 | MN654175 | MN654164 |
| B25PmToKa | Kaminomine/Tokunoshima | *Palythoa mutuki* | Jun 3, 2019 | Reef flat | – | MN654176 | MN654165 |
| B26PmToKa | Kaminomine/Tokunoshima | *Palythoa mutuki* | Jun 3, 2019 | Reef flat | MN654259 | – | MN654166 |
| B28PmToKa | Kaminomine/Tokunoshima | *Palythoa mutuki* | Jun 3, 2019 | Reef flat | – | MN654177 | MN654167 |
| C01PyToKa | Kaminomine/Tokunoshima | *Palythoa* sp. yoron | Jun 2, 2019 | Backreef moat | MN654260 | – | – |
| C02PyToKa | Kaminomine/Tokunoshima | *Palythoa* sp. yoron | Jun 2, 2019 | Backreef moat | MN654261 | – | – |
| C03PyToKa | Kaminomine/Tokunoshima | *Palythoa* sp. yoron | Jun 2, 2019 | Backreef moat | MN654262 | – | – |
| C04PyToKa | Kaminomine/Tokunoshima | *Palythoa* sp. yoron | Jun 2, 2019 | Backreef moat | MN654263 | – | – |
| C05PyToKa | Kaminomine/Tokunoshima | *Palythoa* sp. yoron | Jun 2, 2019 | Backreef moat | MN654264 | – | – |

Mizuyama et al. (2020), *PeerJ*, DOI 10.7717/peerj.8449

Peer J

**Table 1** (*continued*)

| Specimen ID | Location/Region | Spiecies ID | Date (m/d/y) | Environment | Accession no. of ITS | Accession no. of psbA-F | Accession no. of psbA-R |
|---|---|---|---|---|---|---|---|
| C06PyToKa | Kaminomine/Tokunoshima | *Palythoa* sp. yoron | Jun 2, 2019 | Backreef moat | MN654265 | – | – |
| C07PyToKa | Kaminomine/Tokunoshima | *Palythoa* sp. yoron | Jun 2, 2019 | Backreef moat | MN654266 | – | – |
| C08PyToKa | Kaminomine/Tokunoshima | *Palythoa* sp. yoron | Jun 2, 2019 | Backreef moat | MN654267 | – | – |
| C09PyToKa | Kaminomine/Tokunoshima | *Palythoa* sp. yoron | Jun 2, 2019 | Backreef moat | MN654268 | – | – |
| C10PyToKa | Kaminomine/Tokunoshima | *Palythoa* sp. yoron | Jun 2, 2019 | Backreef moat | MN654269 | – | – |
| C11PyToKa | Kaminomine/Tokunoshima | *Palythoa* sp. yoron | Jun 2, 2019 | Backreef moat | MN654270 | – | – |
| C12PyToKa | Kaminomine/Tokunoshima | *Palythoa* sp. yoron | Jun 2, 2019 | Backreef moat | MN654271 | MN654201 | MN654146 |
| C13PyToKa | Kaminomine/Tokunoshima | *Palythoa* sp. yoron | Jun 2, 2019 | Backreef moat | MN654272 | – | – |
| C14PyToKa | Kaminomine/Tokunoshima | *Palythoa* sp. yoron | Jun 2, 2019 | Backreef moat | MN654273 | MN654179 | MN654147 |
| C15PyToKa | Kaminomine/Tokunoshima | *Palythoa* sp. yoron | Jun 2, 2019 | Backreef moat | MN654274 | MN654180 | – |
| C16PyToKa | Kaminomine/Tokunoshima | *Palythoa* sp. yoron | Jun 2, 2019 | Backreef moat | MN654275 | MN654202 | MN654148 |
| C17PyToKa | Kaminomine/Tokunoshima | *Palythoa* sp. yoron | Jun 2, 2019 | Backreef moat | MN654276 | MN654203 | MN654149 |
| C18PyToKa | Kaminomine/Tokunoshima | *Palythoa* sp. yoron | Jun 2, 2019 | Backreef moat | MN654277 | – | MN654168 |
| C19PyToKa | Kaminomine/Tokunoshima | *Palythoa* sp. yoron | Jun 3, 2019 | Backreef moat | MN654278 | – | – |
| C20PyToKa | Kaminomine/Tokunoshima | *Palythoa* sp. yoron | Jun 3, 2019 | Backreef moat | MN654279 | MN654204 | MN654150 |
| C21PyToKa | Kaminomine/Tokunoshima | *Palythoa* sp. yoron | Jun 3, 2019 | Backreef moat | MN654280 | MN654205 | MN654151 |
| C22PyToKa | Kaminomine/Tokunoshima | *Palythoa* sp. yoron | Jun 3, 2019 | Backreef moat | MN654281 | MN654206 | MN654152 |
| C24PyToKa | Kaminomine/Tokunoshima | *Palythoa* sp. yoron | Jun 3, 2019 | Backreef moat | MN654282 | MN654181 | – |
| C25PyToKa | Kaminomine/Tokunoshima | *Palythoa* sp. yoron | Jun 3, 2019 | Backreef moat | MN654283 | MN654196 | MN654153 |
| C26PyToKa | Kaminomine/Tokunoshima | *Palythoa* sp. yoron | Jun 3, 2019 | Backreef moat | MN654284 | – | MN654154 |

**Table 1** (*continued*)

| Specimen ID | Location/Region | Spiecies ID | Date (m/d/y) | Environment | Accession no. of ITS | Accession no. of psbA-F | Accession no. of psbA-R |
|---|---|---|---|---|---|---|---|
| C27PyToKa | Kaminomine/Tokunoshima | *Palythoa* sp. yoron | Jun 3, 2019 | Backreef moat | MN654285 | MN654207 | MN654155 |
| C28PyToKa | Kaminomine/Tokunoshima | *Palythoa* sp. yoron | Jun 3, 2019 | Backreef moat | MN654286 | – | – |
| C29PyToKa | Kaminomine/Tokunoshima | *Palythoa* sp. yoron | Jun 3, 2019 | Backreef moat | MN654287 | MN654208 | MN654156 |
| C30PyToKa | Kaminomine/Tokunoshima | *Palythoa* sp. yoron | Jun 3, 2019 | Backreef moat | MN654288 | MN654182 | MN654157 |
| 159PamToKa | Kaminomine/Tokunoshima | *Palythoa* aff. *mutuki* | July 28, 2010 | In *Mizuyama, Masucci & Reimer (2018)* | MN654300 | – | – |
| 233PamErYa | Yakomo/Okinoerabu | *Palythoa* aff. *mutuki* | Jun 17, 2011 | In *Mizuyama, Masucci & Reimer (2018)* | MN654301 | – | – |
| 237PamErSu | Sumiyoshi/Okinoerabu | *Palythoa* aff. *mutuki* | Jun 18, 2011 | In *Mizuyama, Masucci & Reimer (2018)* | MN654302 | – | – |
| 248PamToKa | Kaminomine/Tokunoshima | *Palythoa* aff. *mutuki* | Jun 21, 2011 | In *Mizuyama, Masucci & Reimer (2018)* | MN654303 | – | – |
| 250PamToKa | Kaminomine/Tokunoshima | *Palythoa* aff. *mutuki* | Jun 21, 2011 | In *Mizuyama, Masucci & Reimer (2018)* | MN654304 | MN654183 | MN654131 |
| 328PamOkTe | Teniya/Okinawa | *Palythoa* aff. *mutuki* | Apr 5, 2012 | In *Mizuyama, Masucci & Reimer (2018)* | MN654305 | – | – |
| 364PamOkOk | Oku/Okinawa | *Palythoa* aff. *mutuki* | Jun 25, 2012 | In *Mizuyama, Masucci & Reimer (2018)* | MN654306 | – | – |

Mizuyama et al. (2020), *PeerJ*, DOI 10.7717/peerj.8449

**Table 1** (*continued*)

| Specimen ID | Location/Region | Spiecies ID | Date (m/d/y) | Environment | Accession no. of ITS | Accession no. of psbA-F | Accession no. of psbA-R |
|---|---|---|---|---|---|---|---|
| 2PtOkOd | Odo/Okinawa | *Palythoa tuberculosa* | Aug 18, 2009 | In *Mizuyama, Masucci & Reimer (2018)* | MN654289 | – | MN654158 |
| 39PtYoUk | Ukachi/Yoron | *Palythoa tuberculosa* | Mar 4, 2010 | In *Mizuyama, Masucci & Reimer (2018)* | MN654290 | – | MN654132 |
| 63PtErYa | Yakomo/Okinoerabu | *Palythoa tuberculosa* | Mar 5, 2010 | In *Mizuyama, Masucci & Reimer (2018)* | MN654291 | – | MN654133 |
| 100PtToKa | Kaminomine/Tokunoshima | *Palythoa tuberculosa* | Mar 9, 2010 | In *Mizuyama, Masucci & Reimer (2018)* | MN654292 | – | MN654128 |
| 15PyOkOd | Odo/Okinawa | *Palythoa* sp. yoron | Sep 5, 2009 | In *Mizuyama, Masucci & Reimer (2018)* | MN654297 | – | MN654130 |
| 51PyYoUk | Ukachi(West)/Yoron | *Palythoa* sp. yoron | Mar 4, 2010 | In *Mizuyama, Masucci & Reimer (2018)* | MN654298 | – | – |
| 85PyErYa | Yakomo/Okinoerabu | *Palythoa* sp. yoron | Mar 5, 2010 | In *Mizuyama, Masucci & Reimer (2018)* | MN654296 | MN654197 | – |
| 105PyToKa | Kaminomine/Tokunoshima | *Palythoa* sp. yoron | Mar 9, 2010 | In *Mizuyama, Masucci & Reimer (2018)* | MN654299 | MN654178 | MN654129 |

Mizuyama et al. (2020), *PeerJ*, DOI 10.7717/peerj.8449

**Table 1** (*continued*)

| Specimen ID | Location/Region | Spiecies ID | Date (m/d/y) | Environment | Accession no. of ITS | Accession no. of psbA-F | Accession no. of psbA-R |
|---|---|---|---|---|---|---|---|
| 218PmOkOd | Odo/Okinawa | *Palythoa mutuki* | May 4, 2011 | In *Mizuyama, Masucci & Reimer (2018)* | MN654294 | – | – |
| 77PmErYa | Yakomo/Okinoerabu | *Palythoa mutuki* | Mar 5, 2010 | In *Mizuyama, Masucci & Reimer (2018)* | MN654293 | – | – |
| 280PmToKa | Kaminomine/Tokunoshima | *Palythoa mutuki* | Oct 5, 2011 | In *Mizuyama, Masucci & Reimer (2018)* | MN654295 | – | – |

## Haplotype network inference and phylogenetic estimation

Obtained sequences for ITS-rDNA, psbA$^{ncr}$ forward and reverse regions were aligned, respectively. In order to discriminate taxa of Symbiodiniaceae, we extracted the ITS2 region utilizing SymPortal (*Hume et al., 2019*; https://symportal.org/) and performed BLASTN search against the nt database using the NCBI website (https://blast.ncbi.nlm.nih.gov/Blast.cgi) for ITS-rDNA sequences. Haplotype network inference was performed for ITS-rDNA sequences using the alignment with TCS networks method (*Clement et al., 2002*) in PopART (*Leigh & Bryant, 2015*). Any columns in the alignment with gaps or ambiguous sites were automatically masked in the inference. The phylogenetic analyses were performed by MEGA version X (*Kumar et al., 2018*) and any loci with ambiguous (double peaks) sites and gaps was automatically deleted completely for calculation in order to avoid over/underestimation of genetic distance among each sequence. Molecular phylogenetic trees of each marker were constructed by maximum likelihood (ML) and neighbor joining (NJ) methods under the JC+G model for ITS-rDNA region and the JC model for psbA$^{ncr}$ regions adopted by modeltest program within MEGA X. The significance of each node was tested by bootstrap test with 1,000 replications. Bayesian inference was performed using BEAST2 (*Bouckaert et al., 2019*) under default settings other than the clock model being changed to the relaxed log normal model, which showed the highest likelihood value according to the model comparison program compiled in BEAST2 (*Drummond et al., 2006*). Posterior probability (PP) on each branch was calculated summarizing four independent 10 million MCMC simulations.

## Statistical analyses

To clarify the relationships between (1) symbiont lineages and host species, and (2) symbiont lineages and host microhabitats, Fisher's exact test was conducted for the compositions of genotype for ITS-rDNA region and monophyletic clades for psbA$^{ncr}$ forward and reverse regions. It should be noted that host microhabitat was restricted by host species for *P.* sp. yoron and *P. mutuki*, and thus we only targeted *P. tuberculosa* for these analyses (aim 2 above) When significance was detected in Fisher's exact test, Cramér's coefficient of association (V) was calculated to evaluate which factors (host species or host microhabitat) were strongly associated with each other.

# RESULTS

## Sequence alignment

The total number of sequences of Symbiodiniaceae from specimens of the four *Palythoa* species obtained in this study was 98 sequences for the ITS-rDNA region (513–773 bp), 40 sequences for the psbA$^{ncr}$ forward region (330–547 bp), and 41 sequences for the psbA$^{ncr}$ reverse region (352–494 bp). As the primer set for psbA$^{ncr}$ used in this study did not make a congruent contig, obtained sequences of forward regions and reverse regions were aligned separately (*Noda et al., 2017*). After alignment, a total of 449 sites with 5 parsimony informative (=PI) sites for the ITS-rDNA region, 260 sites with 94 PI sites for the psbA$^{ncr}$ forward region, and 293 sites with 40 PI sites for the psbA$^{ncr}$ reverse region were used for each phylogenetic estimation.

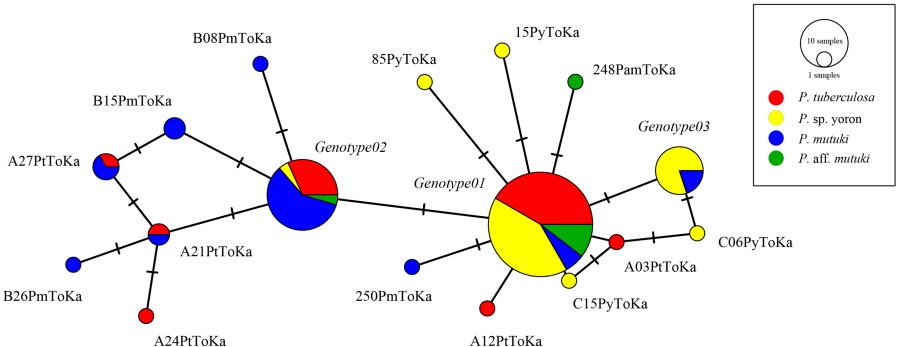

**Figure 3 Haplotype network tree constructed with nuclear ITS-rDNA region alignment using TCS networks method.** Scale represents number of sequences with circle sizes proportional to haplotype frequency. Colors represent *Palythoa* species: red, *P. tuberculosa*; yellow, *P.* sp. yoron; blue, *P. mutuki*; green, *P.* aff. *mutuki*.

## Barcoding, haplotype network and phylogenetic trees

As the result of BLAST searches, all query sequences of the ITS-rDNA region ($n = 98$) were confirmed as belonging to the genus *Cladocopium*. Seventeen ITS-rDNA unique sequences (=genotypes) were observed in TCS network, with most of the sequences belonging to one of major three ITS-rDNA genotypes (Fig. 3, Table S1). No significant clade was detected for the ITS-rDNA phylogenetic tree (Fig. S1). Summarizing these ITS-rDNA genotypes from the viewpoint of host species, *P. tuberculosa* possessed mainly *Genotype01* ($n = 20$) followed by *Genotype02* ($n = 7$), and *P.* sp. yoron also possessed mainly *Genotype01* ($n = 20$) followed by *Genotype03* ($n = 8$) (see details in Table S1). On the other hand, *P. mutuki* possessed mainly *Genotype02* ($n = 13$) with a few *Genotype01* ($n = 3$) and *Genotype03* ($n = 2$). Although the number of specimens examined was smaller ($n = 6$) than those the other species, *P.* aff. *mutuki* also possessed mainly *Genotype01* ($n = 5$).

In contrast, phylogenetic trees generated from psbA$^{ncr}$ regions had a higher resolution. Two monophyletic clades were well supported by bootstrap values and posterior probability in both forward (Fig. 4 *clf1*, ML = 100, NJ = 100, PP = 1 and *clf2*, ML = 100, NJ = 100, PP = 1) and reverse trees (Fig. 5 *clr1* and *clr2*, ML = 100, NJ = 100, PP = 1). Summarizing these Symbiodiniaceae lineages from the viewpoint of host species, *P. tuberculosa* inhabiting the reef edge possessed *clf1/clr1* lineage ($n = 7/5$) and one specimen inhabiting at the backreef moat possessed *clf2/clr2* lineage. *Palythoa* sp. yoron inhabiting at the backreef moat possessed mainly *clf1/clr1* ($n = 9/13$), however, approximately one third of specimens ($n = 5$) possessed other lineages. On the other hand, *P. mutuki* inhabiting the reef flat possessed mainly *clf2/clr2* ($n = 8/8$) other than two specimens that possessed *clf1/clr1*. Unfortunately, as most of *P.* aff. *mutuki* specimens were not amplified by this primer set, we could only obtain phylogenetic information on one specimen which possessed the same lineage as *P.* sp. yoron (C24ToKa-PF) for the forward region and *clr1* for the reverse region.

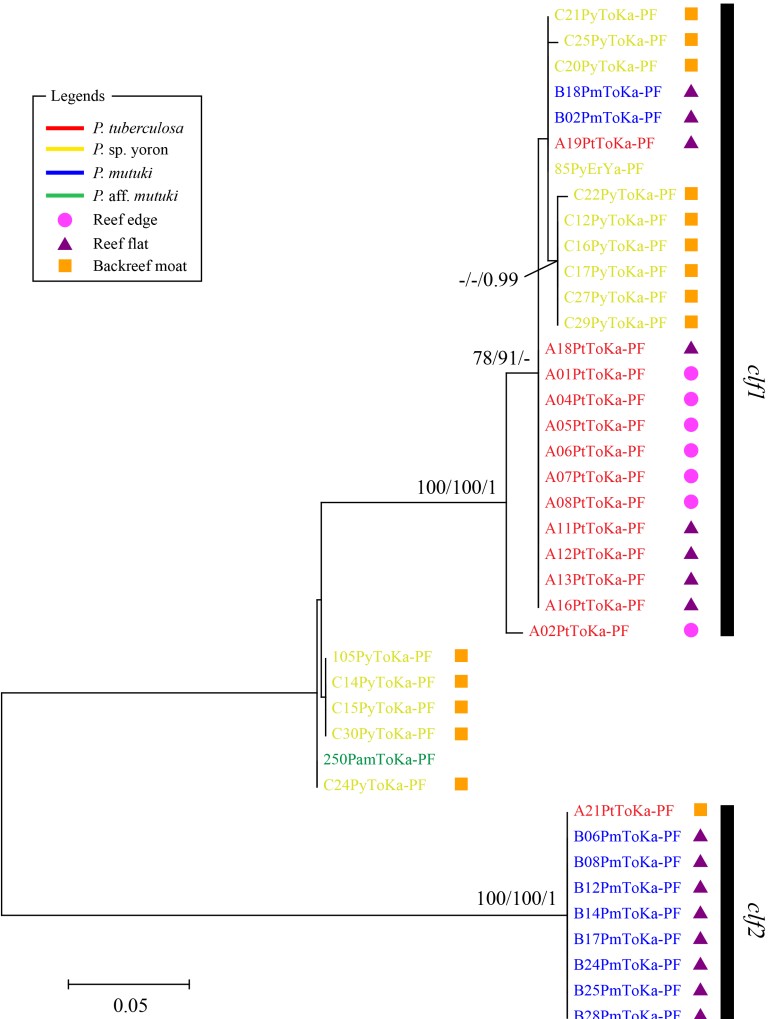

**Figure 4 Molecular phylogenetic tree of Symbiodiniaceae of *Palythoa* species using mitochondrial psbA$^{ncr}$ forward region.** Bootstrap values of maximum likelihood (ML) and neighbor joining (NJ) methods, and posterior probability (PP) are shown more than 70% for ML and NJ, and more than 0.95 for PP at the nodes, respectively. Scale bars indicate substitutions per site. Colored letters and colored diagrams represent *Palythoa* species and their habitats, respectively: red, *P. tuberculosa*; yellow, *P.* sp. yoron; blue, *P. mutuki*; green, *P.* aff. *mutuki*; circle in pink, reef edge; triangle in purple, reef flat; square in orange, backreef moat.

## Relationships among symbiont genotype/lineages, host species and host microhabitats

From the results of Fisher's Exact test, significant differences were detected in all combinations, i.e., ITS-rDNA genotype and host species ($p < 0.01$), psbA$^{ncr}$ forward lineages and host species ($p < 0.01$), psbA$^{ncr}$ reverse lineages and host species ($p < 0.01$), and ITS-rDNA genotype and host microhabitats for *P. tuberculosa* ($p < 0.05$) (Table 2). In other words, it was shown that Symbiodiniaceae lineages and host species were not independent, nor were Symbiodiniaceae lineages and host microhabitats for *P. tuberculosa*.

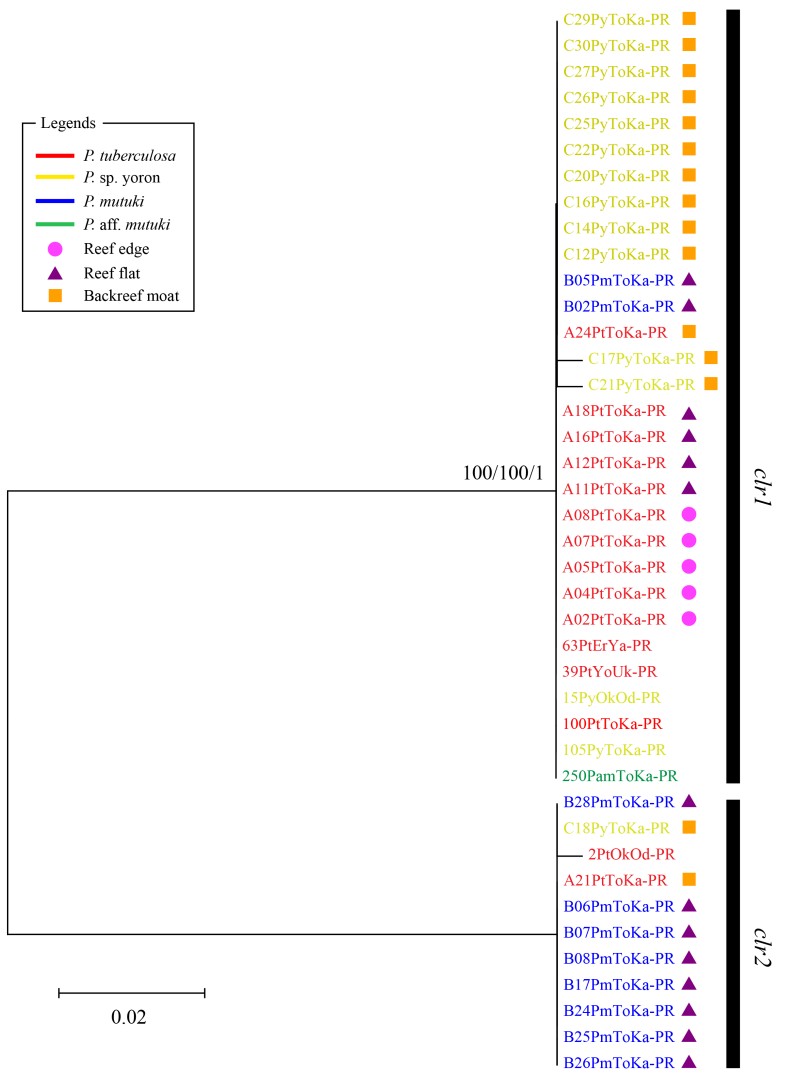

**Figure 5   Molecular phylogenetic tree of Symbiodiniaceae of *Palythoa* species using mitochondrial psbA$^{ncr}$ reverse region.** Bootstrap values of maximum likelihood (ML) and neighbor joining (NJ) methods, and posterior probability (PP) are shown more than 70% for ML and NJ, and more than 0.95 for PP at the nodes, respectively. Scale bars indicate substitutions per site. Colored letters and colored diagrams represent *Palythoa* species and their habitats, respectively: red, *P. tuberculosa*; yellow, *P.* sp. yoron; blue, *P. mutuki*; green, *P.* aff. *mutuki*; circle in pink, reef edge; triangle in purple, reef flat; square in orange, backreef moat.

The effective dose calculated by Cramér's coefficient of association (V) was largest between host species and psbA$^{ncr}$ forward/reverse lineages ($V = 0.786$, $V = 0.682$, respectively), and moderate for the other combinations (host species and ITS-rDNA genotypes, $V = 0.477$; host microhabitats and ITS-rDNA genotypes).

Table 2 **Composition of genotype for ITS-rDNA sequences and monophyletic clades for psbA$^{ncr}$ sequences of Symbiodiniaceae from *Palythoa* species used in this study and microenvironments of host habitats.** Significances were tested by Fisher's Exact Test and V value represents Cramer's coefficient of association.

| | | Symbiodiniaceae genotype (ITS-rDNA) | | | Symbiodiniaceae lineage (psbA$^{ncr}$ forward region) | | Symbiodiniaceae lineage (psbA$^{ncr}$ reverse region) | |
|---|---|---|---|---|---|---|---|---|
| | | *Genotype01* | *Genotype02* | *Genotype03* | *clf1* | *clf2* | *clr1* | *clr2* |
| Host species | *P. tuberculosa* | 20 | 7 | 0 | 13 | 1 | 13 | 2 |
| | *P.* sp. yoron | 20 | 1 | 8 | 10 | 0 | 14 | 1 |
| | *P. mutuki* | 3 | 13 | 2 | 2 | 8 | 2 | 8 |
| | *P.* aff. *mutuki* | 5 | 1 | 0 | – | – | – | – |
| | Total | 48 | 22 | 10 | 25 | 9 | 29 | 11 |
| | | | $p < 0.01$, $V = 0.477$ | | | $p < 0.01$, $V = 0.786$ | | $p < 0.01$, $V = 0.682$ |
| Host habitats of *P. tuberculosa* | Reef edge | 8 | 0 | | | | | |
| | Reef flat | 7 | 2 | | | | | |
| | Backreef moat | 2 | 4 | | | | | |
| | Total | 17 | 6 | | | | | |
| | | $p < 0.05$, $V = 0.508$ | | | | | | |

**Notes.**

*P*. aff. *mutuki* was removed from statistical analyses of psbA$^{ncr}$ region due to low numbers of specimens.

## DISCUSSION

### Symbiodiniaceae genotype/lineage and host species

The development of molecular markers such as psbA$^{ncr}$ that have higher resolution than commonly used 18S or ITS ribosomal DNA markers has helped unveil a more detailed picture of the genetic diversity of Symbiodiniaceae (*Takishita et al., 2003*; *LaJeunesse & Thornhill, 2011*; *LaJeunesse et al., 2018*) (but see also *Hume et al., 2019* who utilized intragenomic variation of ITS2 to resolve genetic delineations). Accordingly, host species biodiversity has been discovered from the initial observation of differences of Symbiodiniaceae phylotypes in some cnidarian species (e.g., gorgonian *Eunicea flexuosa*, *Prada et al., 2014*; scleractinian coral *Seriatopora hystrix*, *Warner, Van Oppen & Willis, 2015*).

From the results of *Mizuyama, Masucci & Reimer (2018)*, none of the four molecular markers utilized could clearly delineate four *Palythoa* species, although they could delineate two closely related species groups composed of *P. tuberculosa—P.* sp. yoron and *P. mutuki—P.* aff. *mutuki*. These previous results seem to be reflected in the results in the current study of Symbiodiniaceae genotypes of ITS-rDNA and lineages of psbA$^{ncr}$ regions. *Palythoa tuberculosa* and *P.* sp. yoron mostly shared the same symbiont genotype (*Genotype01*); nevertheless, they also partially shared the other genotypes with *P. mutuki* (*Genotype02* and *Genotype03*). With regard to psbA$^{ncr}$ lineages, even though the delineation of species groups between *P. tuberculosa—P.* sp. yoron and *P. mutuki* were shown more clearly, they were not divided completely. The situation requires further investigation via obtaining more *P.* aff. *mutuki* specimens' psbA$^{ncr}$ sequences. Unfortunately, in the current study, despite much searching, we could not find large numbers of *P.* aff. *mutuki* on the reef in Tokunoshima Island, even though they were previous sampled for

*Mizuyama, Masucci & Reimer (2018)*. We do not know what happened to *P.* aff. *mutuki* colonies, but they may have been strongly affected by the bleaching events of 2016 and 2017 observed in southern Japan (*Masucci et al., 2019*).

### Symbiodiniaceae genotype/lineage and microhabitat of host species

From the results of the phylogenetic analyses, three microhabitats were not exclusively allocated in distinct Symbiodiniaceae genotypes or monophyletic clades, but the ratios of different genotypes were significantly different for *P. tuberculosa*. Regarding *P. tuberculosa*, Symbiodiniaceae *Genotype01* was mostly detected on the reef edge and reef flat, while *Genotype02* was mainly observed in the backreef moat. Although there were not enough samples to conduct statistical examinations of *P.* sp. yoron and *P. mutuki* due to their habitat specificity, *Genotype02* and *clf2/clr2* were detected mainly on the reef flat, while *Genotype01* and *clf1/clr1* were observed from all three environments.

It has been reported that zoantharian species with different symbiotic genotypes show species-specific photosynthetic responses against seawater temperature and $p$CO$_2$ (*Graham & Sanders, 2016*; *Reimer et al., 2017b*; *Wee, Kurihara & Reimer, 2019*). Although the four *Palythoa* species in this study occurred sympatrically on one reef, the environmental conditions in a reef can be quite different according to small-scale geographical features. Seawater temperatures on reef flats frequently reach near 40 °C (*Achituv & Dubinsky, 1990*). In enclosed reefs, seawater temperatures and $p$CO$_2$ show higher variations than those in exposed reefs (*Suzuki, Nakamori & Kayanne, 1995*; *Fitt et al., 2001*). Thus, the relationship between Symbiodiniaceae and host *Palythoa* species may change among different microhabitats in a reef area, facilitating ecological divergence of *Palythoa* species within a narrow geographic range.

Although a previous molecular study could not distinguish the boundaries among these *Palythoa* species (*Mizuyama, Masucci & Reimer, 2018*), it is suggested by our results that these species are ecologically divergent, and physiological differences within Symbiodiniaceae species may contribute to their ecological adaptation. In fact, *Howells et al. (2012)* reported that *Cladocopium* C1 in *Acropora tenuis* showed different physiological responses between northern and southern populations in the Great Barrier Reef. Considering that *Cladocopium* contains various species distinguished by differences of only a few bp in the ITS2 maker (*Thornhill et al., 2014*), meta-barcoding analyses via next-generation sequencing would be necessary to further understand the detailed relationship between Symbiodiniaceae and *Palythoa* species complex.

## CONCLUSIONS

We succeeded in obtaining genotypic data of Symbiodiniaceae from four putative *Palythoa* species and detected micro-scale geographic variations of the symbiotic algae among these species within a single coral reef. Our results suggest that ecological divergence among *Palythoa* species may be related to differences in Symbiodiniaceae diversities among microhabitats, even within a narrow reef area. More powerful genetic data such as that generated by next-generation sequencing could provide us with additional understanding

on how neighboring *Palythoa* species have co-evolved with Symbiodiniaceae among the different microhabitats in a reef.

### Funding

This study was funded by the research support programs of the National Institute of Advanced Industrial Science and Technology to Masaru Mizuyama. The funders had no role in study design, data collection and analysis, decision to publish, or preparation of the manuscript.

### Grant Disclosures

The following grant information was disclosed by the authors:
National Institute of Advanced Industrial Science and Technology.

### Competing Interests

James D. Reimer is an Academic Editor for PeerJ.

### Author Contributions

- Masaru Mizuyama conceived and designed the experiments, performed the experiments, analyzed the data, prepared figures and/or tables, authored or reviewed drafts of the paper, and approved the final draft.
- Akira Iguchi analyzed the data, prepared figures and/or tables, authored or reviewed drafts of the paper, and approved the final draft.
- Mariko Iijima and Kodai Gibu performed the experiments, authored or reviewed drafts of the paper, and approved the final draft.
- James Davis Reimer conceived and designed the experiments, authored or reviewed drafts of the paper, and approved the final draft.

### Field Study Permissions

The following information was supplied relating to field study approvals (i.e., approving body and any reference numbers):

No permission or institutional approval is required for collecting Zoantharia in Japan. Please see *Mizuyama, Masucci & Reimer, 2018* or contact James D. Reimer for details (University of the Ryukyus; email: jreimer@sci.u-ryukyu.ac.jp).

### DNA Deposition

The following information was supplied regarding the deposition of DNA sequences:

Nuclear internal transcribed spacer ribosomal DNA (ITS-rDNA) and plastid mini-circle non-coding region (psbAncr) sequences of Symbiodiniaceae are available at GenBank: MN654128–MN654306.

### Data Availability

The alignments of sequences for ITS-rDNA, psbA non-coding forward region and reverse region and the chromatogram files of ITS-rDNA region, psbA non-coding forward region and reverse region are available in the Supplemental File. Specimens can be viewed at the National Institute of Advanced Industrial Science and Technology (AIST). A complete list of the specimen accession numbers is available in Table 1.

## Supplemental Information

Supplemental information for this article can be found online at http://dx.doi.org/10.7717/peerj.8449#supplemental-information.

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
