# Peer review of "Comparison of Symbiodiniaceae diversities in different members of a Palythoa species complex (Cnidaria: Anthozoa: Zoantharia)—implications for ecological adaptations to different microhabitats"

_PeerJ, doi:10.7717/peerj.8449_

## Round 0.1 · original submission · Major Revisions

Please provide a point-by-point rebuttal letter to all of the reviewers' comments along with your revised manuscript.

·

Basic reporting

Very good

Experimental design

The design was relevant for the study

Validity of the findings

Off course, other method could provide more resolution

Additional comments

A good work that provide basis for further studies especially the method that will be able to differentiate between species

Reviewer 2 ·

Basic reporting

no comment

Experimental design

The manuscript provide novel and useful information although the purpose is sometimes confusing and not well defined.

Validity of the findings

no comment

Annotated reviews are not available for download in order to protect the identity of reviewers who chose to remain anonymous.

·

Basic reporting

no comment

Experimental design

no comment

Validity of the findings

no comment

Additional comments

Dear Mizuyama et al,

I’ve read your manuscript with interest. You have used nuclear ribosomal DNA and chloroplastic DNA from the psbA to genotype Symbiodiniaceae within a Palythoan species complex. You have statistically shown that the Symbiodiniaceae genotypes are not randomly assorted across the factors of location and host. Rather, you show a host and site driven specificity of the associations.

There is not a lot to go wrong with this dataset and analysis. You have resolved several genotypes with both of the markers and your statistical analyses are sufficient to prove the non-random nature of their distribution.

The writing could improve with regards to the details provided (see specific comments) and in some cases written English. Your analysis of the ITS2 region is far from transparent. You appear to be undertaking editing of the sequences. You also seem to be ignoring additional information in the chromatograms (I cannot comment further because I do not have access to them). You also do not appear to be aware of the amplicon that you are amplifying. You are amplifying the ITS1-5.8s-ITS2 region, not the ITS2 region as this paper is built upon (there may even be 18S and 28S in there, I haven’t checked). This strongly brings into question your analysis of the ITS2 region and the blast results that you provide. I must admit that I find it hard as a reader to trust some of the results with regards to the nrDNA marker. I wonder if better curation of the data would lead to some different results. Unfortunately, I don’t have to time to experiment with this. I would ask that you address some of the specific comments I have raised in the detailed below.

To summarize, your conclusions are largely supported by your results although the nrDNA component of the work should be re-addressed.

I am happy to recommend this work to be published so long as the above issues are thoroughly addressed.

Kind regards,

Ben Hume

Detailed comments
21 The abstract starts with no context of what Symbiodiniaceae are (i.e. zoanthid host-associating algal symbionts), and no context of what “host species” means here (i.e. zoanthid host). The second part of this sentence would be more useful coming first.

44 molecular data, likely due…

81 please provide expected lengths of the amplicons here. You can put what you actually attain in the results.

87 Please provide the chromatogram files as part of the supplementary information so that single peaks can be verified.

87 How and why were the sequences edited?

92 Why are you clustering sequences returned from direct sequencing? This is inappropriate. I am aware of no reason to do this. You should have one sequence per samples and these sequences should be used (after appropriate alignment and cropping for poor quality bp calls) directly for identification of the Symbiodiniaceae. 1bp difference between the most abundant ITS2 sequence represents an enormous difference in terms of phylogenetic and phenotypic distance. As such, the clustering of these sequences will further reduce the resolving power of directly sequenced (i.e. Sanger sequencing) ITS2 amplicon libraries. Perhaps you are undertaking clustering at 100%. Again, without an explanation from you on this, I can’t see what you would do this.

94 above you say that “chromatograms with single peaks were acquired”. Here you talk of how you deal with double peaks. Which is it?

94 The way this is currently written it sounds like you removed only the single nucleotide at which double peaks occurred, or you removed gaps (presumably a gap is from the aligned sequence) in individual sequences rather than removing the entire sequence from the analysis. You cannot modify sequences in this way. You are essentially making up the results by doing this.

96 “each region”. “each marker” would be better.

100 “under default settings other than the clock model being changed to the relaxed log normal model” It would be useful to know why you did this?

112 513-773bp! This is not just the ITS2 region that you are amplifying then. What are you actually amplifying? This will affect the BLAST results too. I’m looking at table 2. It would be useful to see what minimum size match you were using. I.e. if you’re amplicon is 773 bp long, how long are the amplicons that you’re getting matches to. What percentage of the query length are the matches covering?

116 449 sites? Are you saying this is how long the amplicons were? Or rather the number of phylogenetically informative bps?

121 searches

122 Please refrain from using the term “type”. If you mean sequence (which I presume you do), say sequence. Sequence don’t belong to types. Were they exact matches to the C1 sequence? I.e. 0 bp difference? If you mean to refer the the C1 radiation of sequences and species, then say this.

122 “known as generalists in the Indo-Pacific and Atlantic Oceans” This is outdated science. More recent work, including Todd’s work shows that many Symbiodiniaceae are host specific. Modern resolutions also show that the major radiations, i.e. C1, C3, C15, D1, A1 may contains many species, even hundreds (for Cladocopium). See Thornhill 2014 Host specialist lineages dominate the adaptive radiation of reef coral endosymbionts

123 If there were a few substitutions then not all of your samples were matches for the C1 sequence.

124 This is a strange justification; you only need one substitution to make two clades.

125 Genotypes? You’re talking about sequences here right, that represent genotypes.

123 I’ve just looked at your figure 3A. I find it very hard to believe that there was no support for any of these separations. I would like to see the alignment used to make this tree. If there were clear snp differences between the sequences, then these should be seen in the tree.

152 was largest between

171 higher resolution than what?

171 this study is a good example to show that how you analyze the markers is just as important, if not more important than the marker used. If NGS sequencing of the ITS2 had been used and the resulting sequences analyzed to take account of intragenomic variation, it is likely that a resolution equaling that of the psbAncr could have been achieved whilst achieving screening of a wider taxonomic breadth. Smith et al 2017 (Host specificity of Symbiodinium variants revealed by an ITS2 metahaplotype approach), Hume et al 2019 (my own work; SymPortal: a novel analytical framework and platform for coral algal symbiont next‐generation sequencing ITS2 profiling) In fact, it may even be that there is sufficient information in the chromatograms to enable such a resolution. This is currently written with bias towards the psbA marker. COI: I have developed an analytical framework for ITS2; the authors are aware of this from previous reviews.

Fig 3 B and C – why not simply concatenate the fwd and rev psbAncr?

---

## Round 0.2 · Minor Revisions

I truly appreciate the interest of Reviewer 3 for improving your paper. Please pay attention to this last minor comment he has. I totally agree with his comment and I am confident that you will be able to revise fast. As soon as I have your revision, then I can proceed to my final decision.

Reviewer 2 ·

Basic reporting

Although higher number of samples collected in different micro-habitats would be necessary to improved conclusions obtained, authors presented relevant results to future studies.

Experimental design

Aims are well defined compare to the previous version and I think the manuscript has considerably improved.

Validity of the findings

no comment

Additional comments

Authors have addressed all suggestions and comments proposed in the previous version and hence I support it publication in PeerJ

·

Basic reporting

no comment

Experimental design

no comment

Validity of the findings

no comment

Additional comments

Dear Mizuyama et al,

Thank you for taking the time to address the comments made in my review.

You have removed some of the ambiguity in the paper however, there are still some points that could do with improvement (please see below) – some of my points were unaddressed and some are still not clear.

Whilst I believe that the analysis that you have performed could be improved, I see no large issues with the conclusions of the MS as they stand (as I mentioned in my previous review). I would recommend that you have another attempt at addressing the ambiguity and missing information that I point out below, but other than that, I would be happy to recommend this work for publication.

Having taken the time to look through some of your chromatograms I think it is a real shame that you have not taken intragenomic diversity into account when analyzing them. Just from opening up the first three decent chromatograms and looking at them side by side, it is clear that many of the double peaks are non-random (i.e. not sequencing errors; the same double peaks occur in multiple chromatograms). This intragenomic diversity is highly informative! You don’t need NGS to make use it. I’m fully aware that some researchers don’t have access to NGS sequencing, but this is no reason not to make use of intragenomic diversity in your analyses. I have used Sanger sequencing of the full nrDNA region myself (Hume et al.2018: Fine-scale biogeographical boundary delineation and sub-population resolution in the Symbiodinium thermophilum coral symbiont group from the Persian/Arabian Gulf and Gulf of Oman), and my Iranian collaborators who work under exceptionally poorly funded conditions use signals of intragenomic diversity in their chromatograms to identify C. thermophilum and D. trecnchii (e.g. Varaste et al 2018: Symbiodinium thermophilum symbionts in Porites harrisoni and Cyphastrea microphthalma in the northern Persian Gulf, Iran).

I fully appreciate fact that you are pushing to have this paper out in time for a thesis defense and so you are likely uninterested in pursuing this further at this time. However, perhaps, you could mention in your work that this is a viable line of further research that could be taken into account if your analytical approach were to be used again. I think that this would be a good balance to your statements about how lacking the nrDNA marker is in its ability to resolve (in fairness, you appear to have toned down or removed some of these statements now).

I have created an ML tree from your alignment (using iqtree) and indeed I do not see highly supported lineages. Given the ambiguity in your sequence preparation, I would not be willing to say that this is symptomatic of the nrDNA not having the potential to resolve well.

Good luck with the defense!

Kind regards,

Ben

P.s. For future reference, it would be appreciated if you could consistently either mention the line numbers at which you have made your changes (you give them sometimes, but they don’t seem to relate correctly) or include the changes themselves in the rebuttal letter. It’s a little frustrating to have to go through the document looking for the site at which you implemented changes.

101: You are still not being explicit about the region you have amplified. Please use: “partial 18s-ITS1-5.8S-ITS2-partial28S” as you do in your rebuttal letter.

114: Your new text is not helping me understand why you have clustered. I don’t understand how clustering relates to “discriminate taxa of Symbiodiniaceae and obtain representative sequences”. You should not be working with representative sequences; you should be working with the actual sequences. In your rebuttal you say: “We performed clustering to obtain representative sequences (termed as genotypes) to summarize obtained direct sequence data per specimen”. Direct sequence data per specimen is simply the consensus sequence you get from your chromatogram, no? Why would you cluster? In particular you seem to be clustering at 100% identity.

115 You have also not addressed my concerns with regards to what proportion of your sequences you were requiring to be matched when performing your BLASTN. You are using such long sequences and the majority of sequences in the nt database are only the ITS2 region. Are you matching your whole sequence, or only the ITS2, or…?

206. In your rebuttal letter you mention that you are comparing your “higher resolution” to 18S or ITS2 (when direct-sequenced). I don’t see that qualification (direct sequencing) in the new text. You simply flatly compare it to 18S and ITS with no mention of intragenomic variance. In fact now that I look for it, you have not mentioned intragenomic variance once in your entire manuscript despite it being one of the largest complexities in the use of the ITS region.

254. Thank you for including the Hume et al 2019 publication here but it was not my intention to force you to include this. Rather I was hoping to make you more aware of the value of using the intragenomic diversity. This new text currently reads awkwardly. I would suggest you remove the Hume et al reference (which is not the primary literature supporting the point you are trying to make) and replace it with the Thornhill 2014 reference that I suggested in my first review. This paper beautifully points out (using the PsbAncr!) that the ITS2 radiations likely represent multiple (not “various”) species. Importantly, the majority of these species (particularly in Cladocopium) have exactly the same majority ITS2 sequence (i.e. 0bp different). Please readdress this new text accordingly.

---

## Round 0.3 · accepted · Accept

Thank you for the immediate reply to all of the reviewers comments.